# Museformer: Transformer with Fine- and Coarse-Grained Attention for Music Generation

**Botao Yu**[†], **Peiling Lu**[‡], **Rui Wang**[‡], **Wei Hu**[†*], **Xu Tan**[‡*],
**Wei Ye**[§], **Shikun Zhang**[§], **Tao Qin**[‡], **Tie-Yan Liu**[‡]

[†]State Key Laboratory for Novel Software Technology, Nanjing University, China
[‡]Microsoft Research Asia
[§]National Engineering Research Center for Software Engineering, Peking University, China
btyu@foxmail.com, {peil,ruiwa,xuta,taoqin,tyliu}@microsoft.com,
whu@nju.edu.cn, {wye,zhangsk}@pku.edu.cn

## Abstract

Symbolic music generation aims to generate music scores automatically. A recent trend is to use Transformer or its variants in music generation, which is, however, suboptimal, because the full attention cannot efficiently model the typically long music sequences (e.g., over 10,000 tokens), and the existing models have shortcomings in generating musical repetition structures. In this paper, we propose Museformer, a Transformer with a novel fine- and coarse-grained attention for music generation. Specifically, with the fine-grained attention, a token of a specific bar directly attends to all the tokens of the bars that are most relevant to music structures (e.g., the previous 1st, 2nd, 4th and 8th bars, selected via similarity statistics); with the coarse-grained attention, a token only attends to the summarization of the other bars rather than each token of them so as to reduce the computational cost. The advantages are two-fold. First, it can capture both music structure-related correlations via the fine-grained attention, and other contextual information via the coarse-grained attention. Second, it is efficient and can model over $3\times$ longer music sequences compared to its full-attention counterpart. Both objective and subjective experimental results demonstrate its ability to generate long music sequences with high quality and better structures.[1]

## 1 Introduction

Symbolic music generation aims at generating music scores automatically and has drawn more and more attention in recent years [1–3]. Since music can be represented in organized sequences of discrete tokens just like text, Transformer-based models, which have been demonstrated to work well on text generation [4, 5], are increasingly applied in music generation [3, 6–11] and have made great success. While the self-attention mechanism empowers Transformer to capture the complex correlations in music, there are two ubiquitous challenges to solve for this task: 1) *Long sequence modeling*. Music sequences are typically very long, especially for multi-instrument polyphonic music where the lengths can usually exceed 10,000. The quadratic complexity of full attention limits its scalability to that length. 2) *Music structure modeling*. Music has its unique structures, where a piece can usually repeat some patterns of a previous piece, occasionally with some variations, after either a short or a long distance (see Figure 1 for an example). Successfully generating reasonable structures would make the music more realistic just like human-made music.

---

*Wei Hu and Xu Tan are the corresponding authors. This work was partially done while the first author was interning at Microsoft Research Asia.

[1]The generated music samples can be found at `https://ai-muzic.github.io/museformer`. The source code can be found at `https://github.com/microsoft/muzic`.

36th Conference on Neural Information Processing Systems (NeurIPS 2022).

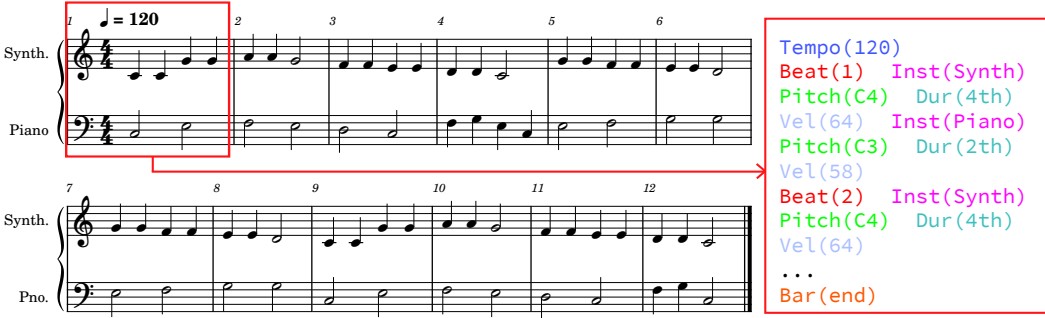

Figure 1: The music score of *Twinkle, Twinkle, Little Star* and its corresponding token representation. Every two consecutive bars on the *Synth.* track have the same rhythmic pattern, and the 9th - 12th bars repeat the 1st - 4th bars with an interval of 8 bars. Structures embodied as repetitions and variations are common in music.

Although many Transformer variants in natural language processing (NLP) [12] have been proposed to handle long sequences (the first challenge), they cannot well model the music structures (the second challenge). According to the basic principles of these models, we can classify them into two types. The first type is called *local focusing*. Models of this type, e.g., Transformer-XL [13] and Longformer [14], mainly focus on part of the input sequence, and drop the rest tokens to reduce the cost. However, the parts that they focus on cannot contain many of the essential ranges important to music structures, and directly dropping the rest tokens may lead to losing some important information. The second type is called *global approximation*. Models of this type such as Linear Transformer [15] utilize linearized attention or sequence compression over the whole input sequence to approximate the token pair-wise attention. While the approximation effectively reduces the complexity, they cannot accurately capture the correlations between related parts and accordingly are inadequate in generating repetition structures. We will review more existing long-sequence Transformers in §2. Directly applying these models to music generation is suboptimal, and it is desirable to design an efficient model that can well model long music sequences as well as their structures.

In this paper, we propose to unify the above two types of models, which can well fit the characteristics of music. Our motivation is based on the observation that the importance is not uniformly distributed over the music sequence, and thus we do not need to treat all the tokens equally. Intuitively, to generate music with repetition structures, the most important information the model should directly refer to when generating a music bar, lies in those bars that tend to be repeated in the current bar. We call these bars *structure-related bars*. For the other bars that are less important, approximation should do the trick. To this end, the proposed fine- and coarse-grained attention takes two different schemes towards different bars: with the fine-grained attention, tokens of the structure-related bars are directly attended to, to well learn the structure-related correlations; with the coarse-grained attention, information of the other bars are captured via summarization [16], rather than attending to each token of them, to decrease the computation and space complexity, and meanwhile retain the necessary information of these bars. The structure-related bars are selected according to the statistics on human-made music. They are not necessarily the most recent and consecutive bars but can include distant ones, to cater to long-term structures in music. Our main contributions in this paper are summarized as follows:

- We propose Museformer, a Transformer model with a novel fine- and coarse-grained attention for music generation. It captures the correlations of structure-related bars via fine-grained attention for learning the music structures, as well as the necessary and concentrated information of the other bars through their summarization via coarse-grained attention.

- We propose to select the structure-related bars based on similarity statistics on human-made music, which can help decrease the perplexity and generate music that exhibits better structures.

- The computation complexity and space complexity are reduced greatly to nearly linear in practice, which enables Museformer to scale up to long music sequences.

- Experimental results show that Museformer can generate music of full-song length with high quality and better structures.

## 2 Related Work

### 2.1 Symbolic Music Generation

Symbolic music generation aims to exploit machines to compose music scores automatically. It has been attracting more and more people to work on it, and the solutions continuously evolve from rule-based models [1] to probabilistic models [17, 18, 2] to deep learning models [6–11].

In recent years, the Transformer-based models have achieved great success in many text generation tasks [4, 5], thus have also been increasingly applied in the similar music generation tasks. Huang et al. [3] apply Transformer in symbolic music generation for the first time and show that it can achieve better performance compared to previous deep learning models such as recurrent neural networks. While Transformer shows promising results in music generation, the quadratic complexity of the attention mechanism limits its applications on the typically long music sequences. To resolve this problem, researchers on symbolic music generation come up with different solutions. One popular solution is to design a new representation method to represent musical information in fewer tokens, such as compound word [19] and OctupleMIDI [20]. While this solution indeed decreases the lengths of music sequences to a certain extent, full-song music sequences are still too long for Transformer to handle. Another solution is to use a long-sequence Transformer variant as the backbone. For example, Huang and Yang [21] use Transformer-XL [13], and Hsiao et al. [19] use Linear Transformer [15]. Although these models can process long sequences, they are initially designed for NLP tasks and cannot well model the music structures. We will introduce and discuss more about the long-sequence Transformers in the following section.

### 2.2 Long-Sequence Transformers

Many Transformer variants have been proposed to tackle long-sequence tasks [12], which in general can be categorized into the following types. 1) Recurrent Transformer, which encodes sequences chunk by chunk [13, 22, 23]. 2) Sparse attention, which sparsifies the attention layout with either predefined patterns [14, 24, 25, 16, 26], or with content-based patterns [27–31]. 3) Linearized attention, which replaces the exponent of the inner product of features with the multiplication of feature maps [15, 32–34]. 4) Compression-based attention, which reduces the number of queries or key-value pairs by compressing the contextual representations [35–37]. In addition, some models also try to combine multiple techniques, such as Compressive Transformer [22] that combines recurrent and compression-based methods, Poolingformer [36] and Transformer-LS [38] that combine sparse attention and compression-based methods.

Existing works on music generation directly adopt some of those long-sequence Transformers to process long music sequences, but it is suboptimal due to the unique structures of music. In general, music has many repeating or similar pieces, many of which are distant from each other, and the distance is measured by time units such as bar or beat instead of the number of tokens. Therefore, the receptive fields of the existing recurrent Transformers or sparse attention methods cannot cover the many ranges of structure-related content. Although the linearized or compression-based attention can cover the whole sequence, they do not precisely capture the correlation between each pair of tokens and may have shortcomings in generating repeating or similar music pieces.

## 3 Museformer

In order to well model long music sequences as well as their music structures, we propose Museformer. The general idea is that we do not need to focus on the whole sequence with the same importance level given that the complexity of pair-wise full attention is unacceptably high, but instead we combine two different attention schemes – fine-grained attention for the structure-related bars, and coarse-grained attention for the other bars. Museformer basically follows the original Transformer architecture [39], and a novel fine- and coarse-grained attention (FC-Attention) is designed to replace the original self-attention module to tackle the challenges in long music sequence modeling.

Museformer takes a music token sequence $X = X_1, \ldots, X_b$ as input, where $b$ denotes the number of music bars, and the $i$-th bar $X_i = x_{i,1}, \ldots, x_{i,|X_i|}$ contains $|X_i|$ tokens. For the $i$-th bar where $i = 1, \ldots, b$, we insert a *summary token* $s_i$ after it to facilitate local information aggregation in FC-Attention. The summary token sequence is denoted by $S = s_1, \ldots, s_b$. After the insertion, the

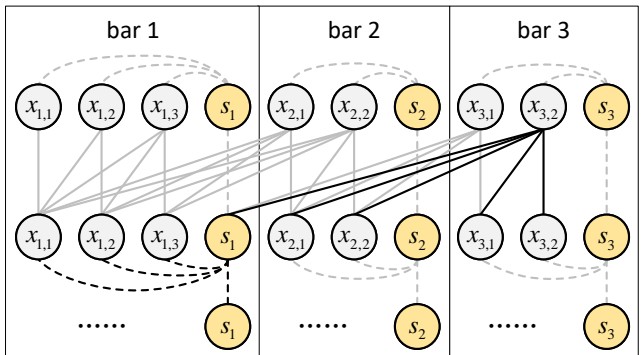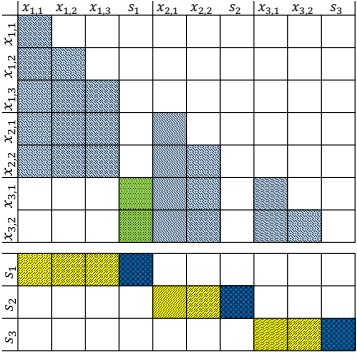

(a) Information flow. The dashed/solid lines depict the summarization/aggregation step. The dark lines represent the information flow towards $x_{3,2}$.

(b) Attention layouts. The bottom/top part corresponds to the summarization/aggregation step.

Figure 2: The fine- and coarse-grained attention. It shows a toy example of 3 bars, where for each bar, only the previous 1st bar is regarded as the structure-related bar.

overall token sequence becomes $X_1, s_1 \cdots, X_b, s_b$. The embedding layer embeds the overall token sequence into a vector space, and concatenates bar embeddings and beat embeddings as the positional information [10], followed by a linear projection. The Museformer layers then model the contextual representation, and the hidden state output by the last layer is fed into a softmax classifier to predict the next token.

In the following part, we first give a brief formulation of the attention mechanism (§3.1). Then, we introduce the details of FC-Attention (§3.2), followed by the selection of the structure-related bars (§3.3). Finally, we discuss the merits of Museformer (§3.4).

## 3.1 Preliminary: Attention

The attention mechanism [39] is the basis of FC-Attention. It receives two sequential inputs, namely source $\boldsymbol{X} \in \mathbb{R}^{n_{\text{src}} \times d}$ and target $\boldsymbol{X}' \in \mathbb{R}^{n_{\text{tgt}} \times d}$, where $n_{\text{src}}$ and $n_{\text{tgt}}$ are the sequence lengths of source and target respectively, $d$ is the embedding dimension. It computes contextual representation for each $\boldsymbol{x}'_i \in \mathbb{R}^{1 \times d}$ in target $\boldsymbol{X}'$ as

$$\text{Attn}(\boldsymbol{x}'_i, \boldsymbol{X}) = \text{softmax}\left(\frac{\boldsymbol{x}'_i \boldsymbol{W}_Q (\boldsymbol{X} \boldsymbol{W}_K)^T}{\sqrt{d}}\right) \boldsymbol{X} \boldsymbol{W}_V, \tag{1}$$

where $\boldsymbol{W}_Q, \boldsymbol{W}_K, \boldsymbol{W}_V \in \mathbb{R}^{d \times d}$ are the trainable parameters. In practice, we employ the multi-head version of attention, but omit it from the equation for simplicity.

## 3.2 Fine- and Coarse-Grained Attention

The basic idea of FC-Attention is that, instead of directly attending to all the tokens which causes the quadratic complexity, a token of a specific bar only directly attends to the structure-related bars that are essential for generating structured music (fine-grained attention), and for the other bars, the token only attends to their summary tokens to obtain concentrated information (coarse-grained attention). To achieve this, we first summarize the local information of each bar through the *summarization* step, and then aggregate the fine-grained and coarse-grained information through the *aggregation* step. Figure 2 visualizes the process.

**Summarization** In the summarization step, we aggregate the information of each bar into the corresponding summary token. For the $i$-th bar, given the representation of summary token $\boldsymbol{s}_i \in \mathbb{R}^{1 \times d}$, and that of the music tokens $\boldsymbol{X}_i = [\boldsymbol{x}_{i,1}, \ldots, \boldsymbol{x}_{i,|X_i|}] \in \mathbb{R}^{|X_i| \times d}$, the summarization of this bar is

$$\tilde{\boldsymbol{s}}_i = \text{Attn}\big(\boldsymbol{s}_i, [\boldsymbol{X}_i, \boldsymbol{s}_i]\big), \tag{2}$$

where $\text{Attn}(\cdot)$ is defined in Equation (1), $[\cdot]$ is the concatenation operation. In this step, each summary token attends to music tokens of its corresponding bar as well as the summary token itself.

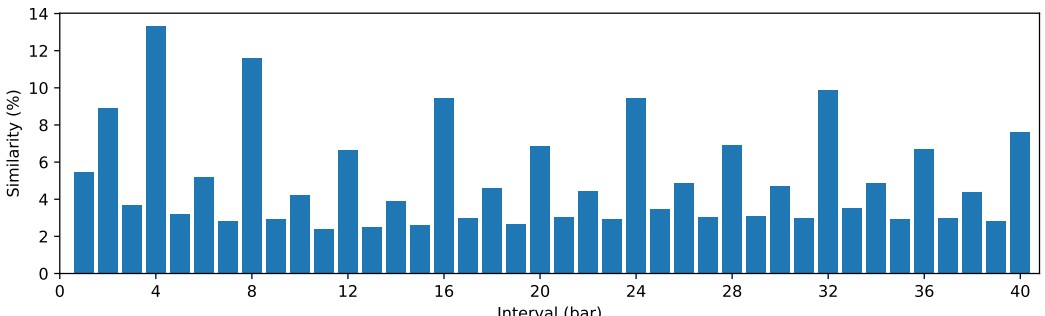

Figure 3: The similarity distribution of the training set used in our experiments. It is calculated over the melody track because melody best shows the structure of a song. We omit the similarity when $t = 0$ because the similarity between a bar and itself is always 1.0.

**Aggregation**   In the aggregation step, we aggregate the information of the tokens belonging to the structure-related bars or within the current bar, as well as the summarization of the other bars, so as to update the contextual representations of music tokens. The updated representation for $x_{i,j}$ is

$$\tilde{\boldsymbol{x}}_{i,j} = \text{Attn}\Big(\boldsymbol{x}_{i,j}, [\boldsymbol{X}_{R(i)}, \boldsymbol{X}_{i,k \leq j}, \tilde{\boldsymbol{S}}_{\bar{R}(i)}]\Big), \tag{3}$$

where $R(i)$ is the set of the indices of the structure-related bars with respect to the $i$-th bar, and $\bar{R}(i)$ is the set of the indices of other previous bars. $\boldsymbol{X}_{R(i)}$ is the matrix formed by stacking $\{\mathbf{X}_i \,|\, i \in R(i)\}$. Similarly, $\tilde{\boldsymbol{S}}_{\bar{R}(i)}$ is the matrix formed by stacking $\{\tilde{\boldsymbol{s}}_i \,|\, i \in \bar{R}(i)\}$. $\boldsymbol{X}_{i,k \leq j}$ is the matrix formed by stacking $\{\boldsymbol{x}_{i,k} \,|\, \boldsymbol{x}_{i,k} \in \boldsymbol{X}_i, \, k \leq j\}$. In other words, the music token $x_{i,j}$ only attends to those music tokens that belong to the structure-related bars, and its previous tokens within the current bar. For the other bars, it only attends to their summary tokens.

### 3.3   Structure-Related Bar Selection

Considering that the structure-related bars are expected to contain those bars that are most likely to be repeated by the current bar to be generated, we propose to pinpoint them by similarity statistics, and the bars with high similarities should be selected. Specifically, for each song in the training set, we calculate the similarity between each pair of the bars, which is defined as

$$l_{i,j} = \frac{|N(i) \cap N(j)|}{|N(i) \cup N(j)|}, \tag{4}$$

where $N(i)$ is the set of music notes within the $i$-th bar, and two notes are considered equal when their pitches, duration, and onset positions within their bars are all the same. The value of $l_{i,j}$ ranges from 0.0 to 1.0. If two bars are exactly the same, this value equals 1.0. We then calculate the average similarity of those bar pairs whose intervals are $t$ over the training set $D$, which is formulated as

$$L_t = \text{Mean}(\sum_{D} \sum_{j=i+t} l_{i,j}). \tag{5}$$

In this paper, we call the distribution of the similarity with respect to the bar intervals the similarity distribution, and show that of the training data in Figure 3. As we can see, it shows an obvious periodical pattern – a music bar tends to be more similar to its previous 2 bars, and also to the previous 4-th bar or its multipliers in most cases. We further conduct the similarity statistics on different datasets involving music of various genres and styles. The results shown in Appendix A interestingly indicate that this pattern is universally applicable to the music of the great diversity. We believe that it can be regarded as a general rule applicable to most music in our daily life.

Based on the statistical results, we carefully select 8 bars – the previous 1st, 2nd, 4th, 8th, 12th, 16th, 24th, and 32nd bars, as the default structured-related bars, since they can cover the most similar bars in most cases. The number of selected structure-related bars is a trade-off between efficiency and information richness. One can select more or other bars according to the computation resources and

the specific similarity distribution of the used dataset. We have to admit that the selected bars cannot always cater to any song, but as long as it covers most cases, it can already enable Museformer to generate music with better structures.

## 3.4 Merits of Museformer

We explain why Museformer is suitable for the music generation task.

First, the receptive fields of the fine-grained attention comply with the characteristics of music and can cover most of the structure-related information. Unlike previous models that adopt fixed token-level patterns [14, 38] or learn content-based patterns [27, 31], which may not cover the most important ranges to be referred to for generating music structures, we directly use the structure-related bars derived from human-made music. These bars can include both neighboring ones and distant ones, which enables Museformer to generate both short-term and long-term music structures.

Second, the coarse-grained attention can preserve necessary information for generating better music. In general, FC-Attention can be categorized into sparse attention, but unlike conventional sparse attention that simply drops a large amount of information, which may limit the model's capability, especially for music where mutual connections abound, the coarse-grained attention preserves the information of the other bars so as to provide rich clues for generation.

Third, the combination of the fine- and coarse-grained attention enables Museformer to handle long music sequences efficiently. Compared to the number of all the music tokens, the number of tokens in the structure-related bars and the summary tokens is much smaller, so the memory consumption and the running time are greatly reduced. It is essential for this task where sequences are quite long.

# 4 Experiments

## 4.1 Experiment Settings

We give a brief introduction to the experiment settings. Please refer to Appendix B for more details.

**Dataset and Music Representation**    We conduct our experiments on the widely used Lakh MIDI (LMD) dataset [40], which contains multi-instrument music in the format of MIDI. After preprocessing, the final dataset contains 29,940 songs (1,727 hours), each song contains 95 bars on average. Following [10], various instruments are merged into the 6 basic ones namely square synthesizer, piano, guitar, string, bass, and drum, with the square synthesizer playing the melody. We use a REMI-like representation method [21] to transfer MIDIs into token sequences, where the musical information (instrument, bar line, note position, pitch, duration, etc.) is represented in separate tokens (see Figure 1 for an example). Each song is represented by 15,042 tokens on average, and each bar is represented by 158 tokens on average. We randomly split all the songs by $8/1/1$ for training/validation/test, respectively.

**Implementation**    Museformer is implemented with PyTorch[2] and fairseq[3]. For the efficient computation of FC-Attention, we write CUDA kernels to construct the attention layouts for each sample, and then transfer the layouts into a blocksparse form and compute with SparTA[4] [41].

**Model and Training Configurations**    The core parameters of Museformer include layer number $= 4$, hidden size $= 512$, number of attention heads $= 8$, and FFN hidden size $= 2,048$. During training, the batch size is set to 4 songs. Following [39], we use the Adam optimizer with $\beta_1 = 0.9$, $\beta_2 = 0.98$ and $\varepsilon = 10^{-9}$, and the learning rate is warmed up over the first 16,000 steps linearly to a peak value of $5 \times 10^{-4}$, and then decreases based on the inverse square root of the steps. The L2 weight decay is set to $0.01$. During inference, we use top-$k$ sampling with $k = 8$. Generation continues until the end-of-sentence token is generated or it reaches the maximum length of 20,480.

---

[2]https://pytorch.org
[3]https://github.com/pytorch/fairseq
[4]https://github.com/microsoft/SparTA

**Compared Models**   We compare Museformer with 4 representative Transformer-based models, most of which have been adopted in music generation:

- **Music Transformer** [3]: a vanilla Transformer with a memory-efficient "skewing" relative position embedding implementation.
- **Transformer-XL** [13]: a recurrent Transformer that encodes the sequence chunk by chunk and uses the gradient-stopped representations of previous chunks as the memory.
- **Longformer** [14]: a model with a sliding window sparse attention.
- **Linear Transformer** [15]: a model that uses a kernel-based attention of linear complexity.

All the compared models are set with comparable hyper-parameters as Museformer. For Music Transformer that uses the full attention and thus cannot model long sequences at once due to memory limit, we chunk each song into multiple samples during training, and apply the model to generate long sequences during validation and inference to test its generalization on long music sequences.

**Objective Evaluation**   We use the following objective metrics to evaluate the models:

- **Perplexity (PPL)**: a common metric to measure whether a generative model can correctly predict future tokens. The smaller, the better. To see the models' performances on different lengths, it is calculated on the first 1,024, 5,120, or 10,240 tokens of each sample.
- **Similarity Error (SE)**: the error between the similarity distribution of training data and generated music, to evaluate the models' ability to generate music with realistic structures. It is defined as

$$\text{SE} = \frac{1}{T} \sum_{t=1}^{T} |\hat{L}_t - L_t|, \tag{6}$$

where $\hat{L}_t$ and $L_t$ are the average similarities (defined in Equation (5)) of the generated music and the training data respectively. We set $T = 40$ in our experiments, and $\hat{L}_t$ is calculated on 100 generated music pieces for each model. The smaller the value is, the more similar the structures of the generated music are to human-made music.

**Subjective Evaluation**   The most canonical way to evaluate a music generation model is the human listening test. We apply each model to randomly generate 100 music pieces, and invite 10 people, where 7 of them have music-related learning experiences, to score these music pieces. Specifically, for each participant, we randomly construct 5 groups, where each group contains 5 music pieces that are generated by Museformer and the 4 compared models, respectively. Participants are asked to score these music pieces from 1 (lowest) to 10 (highest) over the following subjective metrics:

- **Musicality**: whether it is pleasant and interesting, and real enough just like human-made music.
- **Short-term structure**: whether it shows good structures in neighboring content, such as good repetitions and reasonable development.
- **Long-term structure**: whether it shows good structures in long distances, such as song-level repetitions and long-distance connections.
- **Overall**: an overall score. We also calculate **preference score** based on this overall score, which is defined as the ratio of winning times (obtaining the highest overall score within a group).

### 4.2   Comparison with Previous Models

Table 1 shows the results of the objective evaluation. As we can see: 1) Music Transformer achieves a comparable PPL as other models on short music sequences (1,024 tokens) but undergoes a severe deterioration on longer sequences, which implies that a model trained on short music sequences cannot well generalize to long music sequences. Accordingly, an appropriate long-sequence Transformer model is needed for modeling the long music sequences. 2) Although the receptive field of Linear Transformer can cover the whole sequence, its PPLs show no superiority compared to other models, which may be because the kernel-based attention cannot accurately capture the complex correlations of music. 3) The proposed Museformer can consistently achieve the best PPLs on various sequence lengths, especially on a larger length, which demonstrates the effectiveness of Museformer on the

Table 1: The results of objective evaluation and ablation study. The numbers in the parentheses for PPLs are sequence lengths.

|  | PPL (1,024) | PPL (5,120) | PPL (10,240) | SE (%) |
|---|---|---|---|---|
| Music Transformer | 1.66 | 1.77 | 2.55 | 2.49 |
| Transformer-XL | **1.64** | 1.45 | 1.43 | 15.66 |
| Longformer | 1.65 | 1.46 | 1.45 | 5.25 |
| Linear Transformer | 1.86 | 1.67 | 1.64 | 1.97 |
| Museformer (ours) | **1.64** | **1.41** | **1.35** | **0.95** |
| w/o coarse-grained | 1.65 | 1.42 | 1.38 | 1.08 |
| w/o bar selection | 1.65 | 1.43 | 1.39 | 6.39 |

Table 2: The results of subjective evaluation. ST and LT stand for short-term and long-term respectively. For all the subjective metrics, mean $\pm$ standard deviation is reported. Pref stands for preference score.

|  | Musicality | ST structure | LT structure | Overall | Pref |
|---|---|---|---|---|---|
| Music Transformer | $6.00 \pm 2.21$ | $6.90 \pm 1.76$ | $5.30 \pm 2.58$ | $5.90 \pm 1.90$ | 0.20 |
| Transformer-XL | $6.10 \pm 2.19$ | $7.40 \pm 1.81$ | $6.26 \pm 2.78$ | $6.44 \pm 2.01$ | 0.34 |
| Longformer | $6.46 \pm 1.81$ | $7.60 \pm 1.47$ | $6.18 \pm 2.54$ | $6.44 \pm 1.72$ | 0.24 |
| Linear Transformer | $6.06 \pm 1.99$ | $6.92 \pm 2.03$ | $5.78 \pm 2.64$ | $6.30 \pm 1.84$ | 0.24 |
| Museformer (ours) | $\mathbf{6.88 \pm 1.95}$ | $\mathbf{7.86 \pm 1.51}$ | $\mathbf{6.72 \pm 2.74}$ | $\mathbf{7.12 \pm 1.81}$ | **0.46** |

music generation task. 4) The results on SE demonstrate that the music generated by Museformer has structures that are most similar to the human-made music. We provide the similarity distribution for each model and more discussion in Appendix C.

Furthermore, we present the results of the subjective evaluation in Table 2, which show that Museformer achieves the best performance. Specifically, 1) Museformer gets the highest scores on all the metrics. 2) On the structure-related metrics, especially the long-term structure, Museformer exceeds other models by a large gap, indicating that the proposed FC-Attention can empower the model to capture the correlations in distant bars.

We further do pairwise comparisons over the subjective evaluation results. Table 3 shows the number of wins/ties/losses based on the overall scores, as well as the $p$-values of the Wilcoxon signed rank test. Museformer obtains more wins than the compared models, and the $p$-values indicate that Museformer achieves statistically significant improvements ($p < 0.05$).

## 4.3 Ablation Study

We verify the effectiveness of the Museformer components as follows: 1) To see the effectiveness of the coarse-grained attention, we compare Museformer with the setting **w/o coarse-grained**, where each music token attends to the music tokens in the structure-related bars and its previous ones in the current bars, and no summary token of any bar. 2) To see the effectiveness of the structure-related bar

Table 3: The results of pairwise comparisons based on the subjective overall scores.

|  | Wins | Ties | Losses | $p$-value |
|---|---|---|---|---|
| Museformer VS Music Transformer | 33 | 7 | 10 | 0.0003 |
| Museformer VS Transformer-XL | 25 | 14 | 11 | 0.0375 |
| Museformer VS Longformer | 28 | 8 | 14 | 0.0424 |
| Museformer VS Linear Transformer | 29 | 7 | 14 | 0.0128 |

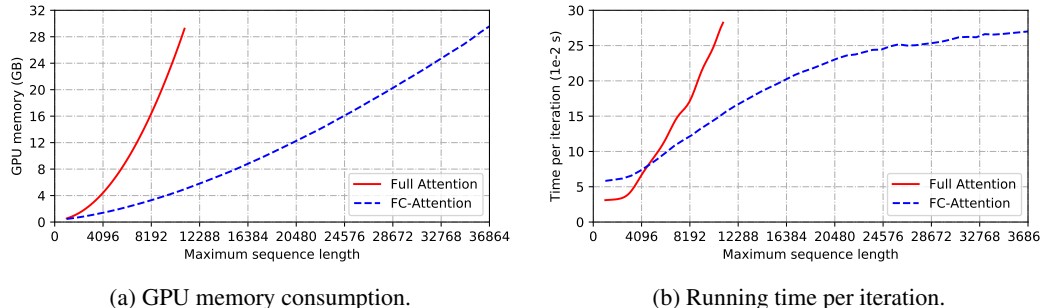

(a) GPU memory consumption.
(b) Running time per iteration.

Figure 4: Memory consumption and running time of Museformer with FC-Attention compared to its full attention counterpart.

selection for the fine-grained attention, we compare Museformer with the setting **w/o bar selection**, where we use the most recent 8 bars for the fine-grained attention instead of the structure-related bars selected via the similarity statistics.

From the results presented in the bottom block of Table 1, we can observe that: 1) Museformer consistently achieves better performances on both PPLs and SE compared to the two ablation settings, which demonstrates the effectiveness of the coarse-grained attention and the structure-related bar selection. 2) The coarse-grained attention preserves the necessary information of the bars other than the structured-related bars and can consistently help decrease the PPLs. 3) The contribution of the structure-related bar selection increases as the sequences get longer. This is reasonable because that in a longer music sequence, there tends to be more bars and more long-term structures, and the fine-grained attention can directly capture the correlations in the distant structure-related bars, which helps make more accurate predictions. 4) Embodied on SE, the bar selection is contributory to the structures of generated music. Please refer to Appendix C for more details. In addition to the above two settings, we have also tried disabling the fine-grained attention for previous bars, where a music token only directly attends to its previous tokens within its bar and attends to all the previous bars via the summary tokens. The result indicates that only summarized information and no precise token-level information for the previous bars is insufficient for the model to generate coherent music.

### 4.4  Complexity Analysis

Suppose the sequence length is $n$, the average bar number is $b$, the average bar length is $m$, and the number of selected structure-related bars is $k$. For FC-Attention, the time complexity of the summarization step is $O(n)$, and that of the aggregation step is $O\big((km + b)n\big)$, so the overall complexity is $O\big((km + b)n\big) = O\big((km + n/m)n\big)$. Although the complexity is still proportional to the square of $n$, the typically large divider $m$ (mostly exceeds 100) greatly reduces the complexity. Thus, the specific complexity is between linear and quadratic. Note that the computation of FC-Attention depends on the input content, i.e., the number of bars and the number of tokens in each individual bar, the complexity cannot be precisely formulated, and the efficiency in real applications is more noteworthy.

To see the efficiency of FC-Attention in real applications, we train Museformer and its full attention counterpart on the validation set, and record their memory consumption and running time. With batch size $= 1$, we increase the maximum sequence length until we use up the 32GB memory of an Nvidia V100 GPU, and record the peak memory consumption[5] and the running time for one-epoch training. Figure 4 shows the results. We observe that the memory consumption of Museformer increases at a nearly linear rate with respect to the maximum sequence length, and can process over $3\times$ longer music sequences than its full attention counterpart, making it capable of generating full-song music. Museformer also achieves faster training time when the maximum sequence length is greater than around $5{,}000$, which is common for music sequences.

---

[5]Obtained by torch.cuda.max_memory_allocated().

### 4.5 Case Study

Figure 5 shows a snippet of a song generated by Museformer. As we can see, on the *Strings* track, the 13th - 16th bars repeat the 9th - 12th bars with an interval of 4 bars, which is a short-term structure. The 25th - 32nd bars repeat the 9th - 16th bars with an interval of 16 bars, and there are reasonable variations in the 27th bar compared to the 11th bar, which shows a long-term structure. This case demonstrates that Museformer can generate music with both short-term and long-term structures, and not only exact repetitions but also some variations. In addition to the two exhibited tracks, other tracks, such as piano and drum, have more variations on the two similar segments, which creates reasonable development of music. Please refer to our demo page for more information.

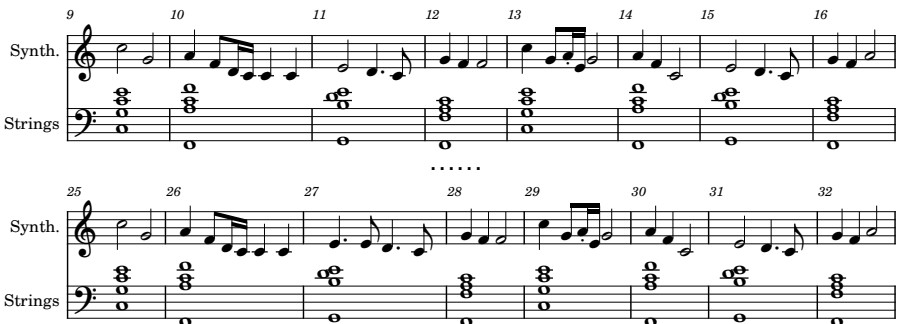

Figure 5: A snippet of a generated song.

## 5 Conclusion

To solve the long sequence modeling and music structure modeling challenges in symbolic music generation, we propose Museformer with a novel fine- and coarse-grained attention. The fine-grained attention is applied to the structure-related bars for learning the structure-related correlations, and the coarse-grained attention is applied to the summarization of the other bars for getting a sketch of them. We propose to select the structure-related bars via bar-pair similarity statistics. Experimental results show that Museformer is efficient and can generate music with good quality and structures.

Museformer is not perfect yet, and we would like to discuss about its limitations and possible future explorations. First, since Museformer takes random samplings during inference and does not receive manual control, it can hardly ensure that every generated music piece is well-structured in an expected way. Techniques to enhance its reliability and controllability can be further explored. Furthermore, the musicality and the creativity of the generated music are still behind those of human-made music, which remains a problem for all the existing music generation models. We believe that more sophisticated music representation and large models trained on large-scale data can help alleviate this problem. Finally, we anticipate Museformer's adaptation to more tasks and domains. It can be easily and reasonably applied to music understanding tasks. For NLP tasks, since natural languages do not have the periodical patterns as music, how to determine the semantically related sentences, paragraphs, or documents is an interesting challenge to solve.

### Acknowledgments

This work is supported by National Natural Science Foundation of China (No. 61872172). We thank the scorers in the subjective evaluation, the SparTA group at Microsoft Research Asia, and many other people from the Websoft group at Nanjing University and Microsoft for their kind help.

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
