# OpenReview forum: "Museformer: Transformer with Fine- and Coarse-Grained Attention for Music Generation"
_NeurIPS.cc/2022/Conference — NeurIPS 2022 Accept_

### Official Review · Reviewer_RLny · 2022-07-08

**Rating:** 4
**Confidence:** 3
**Soundness:** 3 good
**Presentation:** 3 good
**Contribution:** 3 good

**Summary:**

This paper proposed a new transformer-type model that considers long-range musical structures while maintaining linear computational capacity. By separating event-level and bar-level, short-range information is encoded between local tokens while long-range information is encoded in attention in selected bars. As a result, the trained model generated consistent musical structure and achieved better perplexity and human evaluation.

**Questions:**

- Is there any extension of this approach from bars to higher structure, such as motifs, phrases, and so on? If more higher (abstract, sparse) structure does not work, what will be the source of the problem?

- Will the model able to encode musical information that spans longer than 32 measures?


**Limitations:**

- What is key difference between linear attention-type transformers (such as Linformer)? As far as fixing structure of bar-level attention, overall model will effectively work as similar as linear transformer with limited attention view of 32 measures, if data is sufficiently prepared.

**Strengths And Weaknesses:**

Pros
- This work suggested noble transformer architecture considering musical structures.
- Recorded better perplexity and human evaluation compared to previous models.
- Showed consistency in long phrases in audio examples.

Cons
- This approach is not a fundamental improvement of the quadratic attention problem.
- Fits for limited musical structure.

---

> ### Author Response · Authors · 2022-08-02
> **Response to RLny**
>
> We sincerely appreciate you for the valuable comments. In the following response to all the concerns and questions you have posted, we use W for weakness, Q for question, and L for limitation.
>
> **W1: This approach is not a fundamental improvement of the quadratic attention problem.**
>
> Thanks for your question, but we do not really agree that it is not fundamental. On the one hand, our approach is a novel and effective solution to the quadratic attention problem. The comparison in Figure 5 proved that our method significantly increases memory- and time-efficiency. On the other hand, our method is much more than a solution to the quadratic problem, but also a reasonable adaptation to the music characteristics. The ranges for fine-grained attention are decided by the content (the structure-related bars that are not necessarily consecutive) instead of a fixed range of tokens as in many previous works, which enables the model to focus more on the music structures. Also, the FC-Attention enables it to capture both music structure-related correlations via directly attending and other contextual information via summarization, which is a novel multi-granular way. To the best of our knowledge, our method is the first one to model the long-term music structures from the attention perspective, and the experiments showed our superiority on music generation task. Therefore, we believe that our method is a fundamental improvement to the quadratic problem and more importantly can shed light on future exploration on long-sequence music modeling.
>
> **W2: Fits for limited musical structure.**
>
> Actually, the structure-related bar selection strategy used in our paper fits for most of the common-practice music, and we demonstrate it by statistics presented on [this page](https://museformer.github.io/rebuttal): 1) The LMD dataset contains many genres of music other than pop, and we compute the bar-pair similarities for 13 genres labeled by the TopMAGD dataset to see the bar-level structures. 2) We also do the statistics on a [symphony dataset](https://github.com/symphonynet/SymphonyNet).
>
> Also, our method is flexible to all the music structures. For a few genres like freestyle jazz that may not strictly fit for this strategy, the high-level idea that fine-grained attention is for important contents and coarse-grained attention is for other contextual information can be easily adapted to other music structures by simply adjusting the bar selection hyper-parameters.
>
> **Q1: Is there any extension of this approach from bars to higher structure, such as motifs, phrases, and so on? If higher (abstract, sparse) structure does not work, what will be the source of the problem?**
>
> Yes. Our implementation can easily achieve the extension by flexibly setting the ranges for either directly attending or the summary. The reasons why we did not do this extension are: first, bar is the fundamental unit for music, and our bar-level approach can already work well on music generation; second, extension to higher structures would require annotated data, which is expensive and hard to obtain. The extension may work better since more detailed structure-related information is accessible. We leave it for future work. Thank you for the advice.
>
> **Q2: Will the model able to encode musical information that spans longer than 32 measures?**
>
> Yes, through the following two ways: 1) Via coarse-grained attention, other contextual information can be summarized onto the summary tokens and be exploited later. 2) In the multi-layer architecture, information can be passed through the layers, and the union of the attention view on those layers is huge enough to encode a large scope of musical information. An example will be the demo 3 on the [demo page](https://museformer.github.io/). In the video, at 1:33, we annotate that the generated music piece repeats a piece that is 40 bars away.
>
> **L1: What is key difference between linear attention-type transformers (such as Linformer)?**
>
> Our method is quite different from the type you mentioned. Linformer utilizes linear projections to project keys and values from the original sequence length to a smaller length, i.e., compressing the information. The drawback of Linformer is that the input sequence length should be assumed and hence it cannot be used in the generation task. Our method can be in general regarded as a sparse attention. Each token directly attends to a subset of tokens so the complexity can be decreased. However, unlike previous sparse attention works, our method decides the subset according to the data content, i.e., the structure-related bars that are important for the generation for the current bar. Besides, instead of simply dropping other contextual information, our method propose to keep it via the novel bar summary mechanism. The mechanism between the two models are totally different, and our method can achieve the best results compared to the strong baselines.

---

> ### Author Response · Authors · 2022-08-08
> **Response to RLny**
>
> We are wondering whether our last response has properly answered your concerns. If anything is not clear enough, please feel free to ask or have a discussion with us. Thank you very much!

---

### Official Review · Reviewer_Lgcu · 2022-07-11

**Rating:** 7
**Confidence:** 4
**Soundness:** 3 good
**Presentation:** 4 excellent
**Contribution:** 3 good

**Summary:**

This paper presents Museformer, a music-inspired transformer architecture which uses fine-grained and coarse-grained attention to be able to model very long sequences of music while being able to maintain structure-awareness. The proposed architecture reduces the memory and computation cost from quadratic in standard transformers to linear which helps with scaling to long sequences. The transformer architecture involves learning at individual note-level as well as bar-level. The authors determine so-called structure related bars for every bar in an input piece. These are determined using prior knowledge of music and are chosen as the previous 2 bars (for short-term context) along with previous bars at a distance of 4, 8, 16, 24 and 32. This is obtained by analyzing the corpus which is a pop music corpus and most songs use standard 4/4 rhythms. The fine-grained attention attends to each token of all structure-related bars, whereas the coarse-grained attention attends to a summary token for every other previous bar. Finally a combination step combines the information from the current bar, the fine-grained attention, and the coarse-grained attention using another attention layer.

The evaluation includes comparison against various Transformers for long sequences. The authors use Perplexity as an objective metric and also conduct a small scale user study with 10 participants. The Museformer architecture outperforms all other baselines on all metrics. An ablation study also shows the benefits of coarse-grained attention and structure-related bar selection. The authors also demonstrate the time and space complexity benefits against vanilla transformers for varying sequence lengths.

**Questions:**

- Why do the authors choose to make the model causal during training? I get that the inference process has to be causal, but during training wouldn't it be better to also attend to future bars?
- Please see weakness section for question about evaluation of generated structure.

**Limitations:**

The authors briefly talk about the complex implementation of dynamic sparse transformers as being a limitation. I would like the authors to also possibly discuss some limitations of the generated results; for example, I find that the drums were very unnatural. I'm sure the authors also have some thoughts about the generation quality which readers might be interested to hear about.

**Strengths And Weaknesses:**

Strengths:
The museformer architecture uses domain-knowledge of music to reduce the complexity of Transformers to be able to generate long sequences of symbolic music. There are other papers that attempt to address the issue of maintaining long-term structure in music, but most attempts are unable to consistently maintain long-term structure. The demo samples from the authors demonstrate repetitions at large gaps as well as short term repetitions. While I do find the samples to still lack quality, the work is a step in the right direction. Additionally, the authors choose strong baselines for long-sequence learning and conduct the relevant ablation studies to show the benefits of their components.

Weaknesses:
I find the evaluation to be a little weak. The authors focus on long-term structure but do not perform any objective comparison of the generated structures to structures from real music. Some kind of self similarity matrix comparison might be interesting to look at. Some statistical analysis as well. I get the feeling that the museformer really focuses on producing almost exact copies of structure-related bars and real music might have more variation. Some visual comparison of these SSMs might be interesting. The writing could also be improved a bit but that is a minor weakness. There are a few phrasing issues.

comment:
L231: please fix the commas in the numbers. This took me a while to figure out that perplexity is measured over 1024, 5120, and 10240 tokens. I got it after looking at the tables but was very confused at first glance.

---

> ### Author Response · Authors · 2022-08-02
> **Response to Lgcu**
>
> We sincerely appreciate you for the valuable comments. In the following response to all the concerns and questions you have posted, we use W for weakness, Q for question, and L for limitation.
>
> **W1, Q2: No objective comparison about structures is performed. Some kind of SSMs as well as some statistical analysis might be interesting to look at.**
>
> It is difficult to design an objective metric that comprehensively reflects the music structures, in light of their diversity. Detecting music structures is still being explored. Accordingly, we demonstrate our ability to generate music structures in the subjective evaluation with the metrics namely short-term structure and long-term structure. According to your advice, we have calculated the SSMs, which can be found at [this page](http://museformer.github.io/rebuttal).
>
> **W2: Museformer may produce almost exact copies instead of more variation as in the real music. Some visual comparison of these SSMs might be interesting.**
>
> Sorry for leaving you the feeling, which may be because we emphasize repetitions in our text and demos, yet Museformer does not simply copy. For example, in the video of demo 1 on the [demo page](https://museformer.github.io/), the piece at 1:14 repeats the melody at 0:29 with some reasonable variations, and the two pieces are completely different on the accompaniment tracks, which shows the development of music and is quite realistic as human-made music. We will present their SSMs on [this page](http://museformer.github.io/rebuttal).
>
> **W3, W4: The writing could be improved, and comma problem in Line 231.**
>
> Thanks for pointing it out. We have fixed the comma problem and refined the whole paper.
>
> **Q1: Why causal during training?**
>
> For a generation model like Museformer, to function well in inference, it needs to be trained in the same setting as in the inference process, i.e., only attending to previous tokens. Other generation models such as GPT are also trained in this way.
>
> **L1: More discussion about limitation of the generated results can be added.**
>
> The quality of generated music is always the key challenge that all data-driven methods want to address. The difficulties may come from the diversity of music like the different textures or genres. Some work (e.g., Music Transformer) generates classical music with only piano tracks. Other work aims at generating multi-instrument accompaniments (e.g., PopMAG). The results from these works are not always as realistic as human-composed ones since music is built up by a combination of rules and creativity. Previous work did a good job of generating short music segments, but in real composition, music ideas are organized and emphasized by music structures. Thus, to generate more realistic music, our work proposes this fine- and coarse-grained attention mechanism to generate long-sequence music with music structures with repetition and variation. The subjective results also demonstrate the good musicality and reasonable structures of the generated samples compared to baselines. However, we must admit there is still a long way to go to generate music that is identical to human-made ones, but we will keep investigating more effective ways to help boost this research. Thank you for the advice. We will add the discussion to the paper.

---

> ### Author Response · Authors · 2022-08-08
> **Response to Lgcu**
>
> We are wondering whether our response has properly answered your concerns. If anything is not clear enough, please feel free to ask or have a discussion with us. Thank you very much!

---

> ### Comment · Reviewer_Lgcu · 2022-08-08
> **Response to Authors**
>
> I thank the authors for their detailed response. I am satisfied with the responses to my questions and comments about the weaknesses of the paper. I looked at the SSMs on the demo page. Thanks for generating these, however, I was also looking for comparison with real music. Also did you generate the SSMs at the instrument track level, or for the song as a whole? I would assume that most of the variation would occur in the melody track, while the rhythm and harmony/chords tracks are essentially repeating throughout with perhaps a key change at some point as is the case in most pop music.

---

> > ### Author Response · Authors · 2022-08-09
> > **Response to reviewer Lgcu**
> >
> > Thanks for your response.
> >
> > For the comparison with real music, we have presented the similarity distribution of all the training samples at the bottom of [this part](https://museformer.github.io/rebuttal#music_analysis). If we compare it with the distribution of the demos generated by the models, we can see that our method (Museformer) is the closest to the real music. Also, we uploaded just now some more SSMs of training samples FYI, which you can find [here](https://museformer.github.io/rebuttal#ssm_train). The demos generated by Museformer in general manifests similar characteristics of the real music.
> >
> > As for the tracks, all the statistics and the SSMs you have seen are done only on the melody track, because we think melody is the main information of music and other accompaniment tracks are auxiliary to melody when it comes to music structures. We also uploaded the SSMs of the other tracks, and you may check them by clicking the “SHOW OTHER TRACKS” button in [this section](https://museformer.github.io/rebuttal#ssm). As you assume, in general, the melody shows more variations, while the accompaniment tracks usually follow a specific pattern and repeat more. This is consistent with our experiences in music.
> >
> > Please feel free to contact us if there is anything that is not clear or you are curious about. Thanks!

---

### Official Review · Reviewer_kVqg · 2022-07-11

**Rating:** 7
**Confidence:** 3
**Soundness:** 3 good
**Presentation:** 3 good
**Contribution:** 3 good

**Summary:**

The paper proposes a form of structured, sparse self-attention designed to work well for symbolic pop music generation
with transformer networks.  Instead of uniformly attending to all previous tokens in a sequence, the proposed method
reduces computation by only attending to 1) per-bar summary tokens (coarse attention) and 2) full-token sequences _only_
within a small fixed set of "structure-related bars" (sparse fine-grained attention).

This structure incorporates domain-specific knowledge (and empirical analysis) of common musical structure to identify
the set of "structure-related bars" and results in high quality generation with 1) reduced computation compared to
full-context attention models and 2) improved long-term structure compard to several baselines, including long-term
repetitions/recapitulations commonly found in western pop music.


**Questions:**

- How are bar lines determined?  Are bar boundaries explicitly represented in all songs of the Lakh MIDI dataset, or do
  they need to be inferred somehow?

- Sec 3.2, lines 159-166: Would this distribution differ in other corpora or genres?  Seems like enforcing such a
  constrained bar relation pattern would make it more difficult to model music with unusual structures, a limitation of
  the proposed approach.
  - Also, according to Sec A.1, only 4/4 music is considered.  Is the proposed structure not suitable for other
    time-signatures?  Was time-signature generalization considered empirically at all?

- line 172: $\bar{N(i)}$ notation is not defined.  Does this simply indicate non-structure bars?

- line 187: It seems that this method could easily be be considered a form of sparse attention with a predefined
  structure?

- line 193: how much smaller is "much smaller"? I'd expect an average bar length of something like 8 or 16 notes?  line
  202 claims 158 tokens per bar on average.  Why is this so much higher?  It would be helpful to future readers to
  elaborate on this.  Presumably it is related to the REMI encoding shown in Fig 1 which uses 5+ tokens per note
  (separating the note identity, pitch, velocity, duration, instrument, etc)?

- Sec 4.4. lines 279-281: The claim that FC-attention is linear feels a bit overstated. The number of bars $b$ will
  scale linearly with $n$ since bar lengths are generally independent of overall sequence length, i.e., $b = n / K$
  assuming K token per bar.  So the overall complexity would be $O(b n) = O(n^2/K) = O(n^2)$, remaining quadratic in
  sequence length.  Due to the large discount factor $K$ it is clearly much sparser than full attention, but the
  *asymptotic* complexity remains quadratic even if it is reduced by a large factor.

- Fig 5: Why aren't other methods included in this figure?  E.g., transformer XL?

**Limitations:**

As described above, it would be helpful for the authors to address the limitations of the proposed approach across musical genres, time-signatures, etc.  Is the set of  "structure-related bars" used in the experiments thought to be generally applicable across musical corpora, or should they be empirically determined using an analysis similar to the "Structure-Related Bar Selection" procedure on a dataset-by-dataset basis?

Furthermore, any exposition on the potential applicability of similar ideas to other domains, e.g. long-form text generation, would be interesting to see.

**Strengths And Weaknesses:**

Strengths
- Straightfoward, well-motivated approach, works well on a challenging area of generating long sequences (10k+ tokens)
  with coherent long-form musical structure.
- Well rounded evaluation, including nice demos.  I especially appreciate the inclusion of subjective evaluations given
  the lack of objective metrics for realistic musical structure.
- The text is well written and clear.

Weaknesses
- Very specific application in music, perhaps somewhat narrowly at that (see below).
  - It would be most interesting to hear whether the authors imagine similar ideas applying to NLP, e.g., by incorporating
    sentence-level summary tokens to cheaply expand the potential context and/or explicitly attending to, say, the first
    and last sentence of nearby paragraphs, analogous to the proposed structure-related bars.
- The repeated characterization of the proposed method reducing the computation complexity of attention from quadratic
  to linear (e.g., lines 69-71 and elsewhere in the text) is, at best, overstated.  Since the ratio of bars to tokens is
  a roughly fixed proportion, the proposed model still requires quadratic attention across bar-summary tokens, albeit
  scaled down by a large factor corresponding to the average number of tokens per bar (~150 according to the paper).
  The text should be updated to clarify this claim.

---

> ### Author Response · Authors · 2022-08-02
> **Response to kVqg**
>
> We sincerely appreciate you for the valuable comments. In the following response to all the concerns and questions you have posted, we use W for weakness, Q for question, and L for limitation.
>
> **W1, L2: It is very specific application in music, perhaps it can be applied in NLP.**
>
> Indeed, Museformer is proposed for music, so its design can reasonably fit the music structures.
>
> However, the general idea of Museformer, that to combine fine-grained attention for important content and coarse-grained attention for other contextual information, is universal to all kinds of sequence data, especially the hierarchical sequence data (e.g., music: song-bar, text: document-paragraph-sentence), and thus we really have considered its application on NLP. However, we also need to consider the difference that natural language is not as structural as music, i.e., it does not regularly repeat previous content as in music, and thus we cannot tell that which sentence should be directly attended according to only the position. Therefore, we should find a way to correctly select semantically important sentences when applying Museformer in NLP. Your suggustion that to use the first and last sentence in the near paragraph really offers an insight to bridge the semantic meaning and the position. In fact, we have been working on some selective attention mechanism to automatically select the important part, which we leave for future work. According to your advice, we will add the discussion in the paper.
>
> **W2, Q6: The complexity of attention.**
>
> Thanks for pointing it out, and we have clarified it in the paper. In real scenarios, the bar length is usually large, so the complexity should less than square and larger than linear.
>
> **Q1: How are bar lines determined?**
>
> The MIDI files do not contain bar boundaries, and we infer them with the time signature information contained in the MIDI files. For example, if the time signature is 4/4, then a bar line should lay between the 4th beat and the 5th beat.
>
> **Q2, L1: Can structure-related bar selection generalize to other corpora or genres, or to other time signatures? Or should they be empirically determined using a dataset analysis?**
>
> Yes. We have done a bar-pair similarity statistics over 13 genres of music on the LMD dataset, and over a new symphony dataset. The results (can be found [here](http://museformer.github.io/rebuttal)) show that this bar selection strategy is applicable to most music. The datasets also involve many time signatures, and the time signature has no strong relation with music structure, so this strategy can naturally generalize to other time signatures.
>
> However, we must admit that there exist some genres like freestyle jazz that are not strictly fits for this music structure. For those genres, conducting an empirical dataset analysis such as computing the bar-pair similarities is a good way to decide the strategy, and our model can be easily adapted to new strategy by simply setting the bar selection hyper-parameters.
>
> **Q3: $\bar{N}(i)$ in line 172 is not defined, and does it indicate non-structure bars?**
>
> Exactly, and it is defined in line 171.
>
> **Q4: In line 187, Could this method be considered a form of sparse attention with a predefined structure?**
>
> In general, yes, because we do attention computation over a subset of tokens. However, unlike the previous sparse attention methods (sliding window, random sparse pattern, etc.) that directly drop the information, we propose to use the bar-level summary to hold the information, and the ablation study proved its effectiveness.
>
> **Q5: How much smaller is “much smaller” in line 193. Why is the average number of tokens per bar so high?**
>
> Suppose the sequence length is $n$, and the average length of each bar is $m$, then the number of bars $b$ should be $n / m$, which is also equal to the number of summary tokens. In real application, $n$ can be easily larger than 10k, while $b$ is usually around 100. The reasons why the average number of tokens per bar ($m$) is so high are as follows: First, music contains pitch, duration, velocity and many other meta information, which is represented in separate tokens. Second, there are many instruments’ tracks and many notes in multi-track polyphony music. We will add more details about it in paper. Thanks for your advice.
>
> **Q7: Why aren’t other methods included in Figure 5 (e.g., Transformer-XL)?**
>
> Here in Figure 5, we only aim to show the efficiency of the FC-Attention compared to the conventional full attention, and thus demonstrate our ability to efficiently process long music sequences. Also, the fact that different models differ in components other than attention module, as well as their implementation, would not let the comparison be fair enough. All the baseline methods except for Music Transformer should be able to efficiently process long sequences, while Museformer is designed for the music structures and the experiments demonstrate our superiority.

---

### Official Review · Reviewer_MUNC · 2022-07-11

**Rating:** 4
**Confidence:** 5
**Soundness:** 2 fair
**Presentation:** 1 poor
**Contribution:** 2 fair

**Summary:**

The paper proposes Museformer, a Transformer-based music generation model which can handle long music sequences via tailored design of attention mechanism, called Fine- and coarse-grained attention. Specifically, the long music sequences are broken into bars and attention are designed to connect with both local tokens and structurally-important bar summary tokens. Results and analysis show the model perform better in long sequence generation compared to existing Transformer-based models and computes attention in linear time complexity.

**Questions:**

1. The major concern for this paper is about the novelty of the proposed method. Music structure is a hard problem because simple assumptions fail. I believe it is okay to simplify the assumption to a certain extent, as the authors made in this paper. However, it needs to be proved in the experiment where the method can be generalized to a representative dataset. Lakh MIDI dataset shares the same assumption with POP909 so it is not proper. But it could also be the case that I did not get it. So I would like for more explanation why the proposed paper is not a trick but a systematic method for music.

Below are the questions related to experiments:

2. In table 2 and the relevant text, what test is used and how is p-value computed?

3. If I understand it correctly, in Table 3, "Museformer w/o coarse-grained attention" is almost the same as "Music Transformer". (They might be different in the implementation of bar summary.) So, why is the PPL so different for Music Transformer and "Museformer w/o coarse-grained attention"? Should we conclude "bar summary" is a more fundamental improvement compared to FC-attention?

Below are the questions related to demo:

4. Lakh MIDI dataset has multiple tracks, however in each track, the music is relatively simple (monophonic or chords with simple texture). How is the model on a dataset where we have less track but more complicated? (E.g., a dataset like Maestro)

5. The demos have very different instrumentation and beginning. Are all these samples generated from scratch?

6. The demos have clear repeats and actually sometimes too many repeats. However, grasping the whole song structure and keeping the song interesting is still difficult. I wonder if the claim for music structure is too strong. Also, I noticed the melody is not so human-like compared to existing music generation algorithms.

**Limitations:**

Limitation of this paper is discussed in the weakness section and question section. My suggestion is to consider claim the model to achieve a weaker goal, such as, a music generation algorithms for pop music which consider structural repetition.

**Strengths And Weaknesses:**

Strengths:
* Although there are many studies on automatic music generation, few study consider the music generation under the scope of the whole song. This work proposes an adaptation of the powerful Transformer model to the music domain.
* The assumption of the paper (the subset of music structure considered) is very clear and addressed and evaluated throughout the paper.
* The adaptation to music is both scientifically efficient and makes some sense in music.
* implementation detail is clear.

Weakness:
Music structure is a difficult problem. The main weakness of this paper is some too strong assumptions made in this paper about music structure (e.g., long-term structure is regarded as 2, 4, ..., 32 bars ahead). Such assumption cannot be tested in the paper because the model is evaluated on a dataset where such assumption prevails. In other words, such assumption is not general for music. It is a novel *trick* for certain styles of music, however it remains to be tested whether it is a *method* (for all music styles.
* Although FC-attention is powerful in the given experiment setting, the method is actually quite not novel. In a nutshell, if attention can be done to a subset of *fixed number of tokens*, the computation is of course done in linear complexity. The fundamental hard problem is how to determine a subset. However, in this paper, it assumes the subset is the multiples-of-two-bars ahead. Such estimate can be generally true for pop music, but definitely not true for other styles. The dataset for the estimate is also quite arbitrary.
* I am skeptical about Table 1 - Table 3. Please see the questions below.
* Figure 6 is not well-formatted for score reading. Maybe there are better examples, but the current one is not musical enough by just reading this example. Please note that repetition is not always good. Rigid repetition can be bad.
* Demos are in general not musical enough. I have listened to the proposed demo and have some concern. Please see the questions below.

Other issues:
* The method is clear but the writing is a bit wordy and sometimes explain things illogically. I have to understand the meaning with empathy. For example,
    * ll. 112-115 is illogical. The first sentence is not the cause of the second sentence.
    * ll. 146-150 is illogical. The clause after "instead of" is only part of the "basic idea of FC-Attention", as it only discusses fine-grained attention.
    * ll. 134-136: what is $d$?
    * ll. 159 - 164: "music tends to repeat every 4 bars..." is not true from the previous experiment. It is true that "music from the POP909 dataset tends to repeat every 4 bars".

---

> ### Author Response · Authors · 2022-08-02
> **Response to MUNC**
>
> We sincerely appreciate you for the valuable comments. In the following response to all the concerns and questions you have posted, we use W for weakness, Q for question, and L for limitation.
>
> **W1, Q1: The assumption about music structure may not be general for all music styles.**
>
> The structure-related bar pattern used in our paper should generalize to most common music. We compute the bar-pair similarity distributions with respect to the distance over 13 genres of music on LMD and another symphony dataset, and the results show the pattern consistently holds for these genres and datasets, which suggests that the pattern is widely used and relatively universal. More details can be found on [this page](https://museformer.github.io/rebuttal).
>
> We must admit that there exist some genres like freestyle jazz that do not strictly fit for this strategy. However, the high-level idea that fine-grained attention is for important contents and coarse-grained attention is for other contextual information can be easily adapted to other music structures by simply adjusting the bar selection hyper-parameters.
>
> We truly recognize that, to solve the difficult music structure problem, there is still a long way to go. To the best of our knowledge, our work is the first to solve this problem on the full-song level and from the attention perspective. We believe our work can shed light to future exploration on this problem.
>
> **W3: Figure 6 is not well-formatted. The example is not musical enough.**
>
> Thank you for pointing it out. We have updated the format of the case.
>
> This case is only an excerpt of the accompaniment piano track, which demonstrates our ability to generate both short-term and long-term structures. Only by just reading the limited excerpt may be not enough to tell the musicality. We suggest that you may refer to its complete version (i.e., demo 1 on the demo page) to evaluate its musicality.
>
> Although we show in this case only repetitions, our generated music does not repeat rigidly. As shown in the video of demo 1, the piece at 1:14 repeats the melody at 0:29, yet with reasonable variations, and the two pieces are completely different on the accompaniment tracks.
>
> **W4, Q6: Demos are not musical enough.**
>
> Compared to the existing methods, our setting is different and more difficult. 1) We focus on modeling music structures on the full-song level. The experiments show that our method can generate songs with reasonable structures and also good musicality compared to the strong baselines. 2) The dataset we use is a multi-track polyphony dataset, while many previous works use datasets with only a piano track (e.g., Maestro, Pop1k7). Modeling the interplay among multiple instruments can be much harder. We will upload more demos to show the musicality.
>
> **W5: The writing is a bit wordy and sometimes explains things illogically.**
>
> Thanks for pointing out these problems. We have fixed the illogicality and refined the whole paper.
>
> **Q2, W2: In table 2 and the relevant text, what test is used and how is p-value computed?**
>
> Following Pop Music Transformer (Huang et. al., 2020), we use the Wilcoxon signed rank test, and compare Museformer with each of the baseline models based on the overall scores to calculate the p-value, which is described in detail in the Appendix C.
>
> **Q3, W2: Are "Museformer w/o coarse-grained attention" and Music Transformer almost the same? If so, why PPLs are so different? Is "bar summary" more fundamental?**
>
> No, they are different. “Museformer w/o coarse-grained attention” (MFwoCA) means removing the bar summary and only keeping the fine-grained attention, while Music Transformer (MT) attends to all of the previous tokens. As you may see, MFwoCA achieves better PPL than MT when length is 1024, which demonstrates the effectiveness of fine-grained attention. That the PPLs are so different is because of the training setting (line 224-227): MT uses a full attention and cannot process a long sequence at once, so following its paper, we chunked the sequences during training. When the length increases, the PPL of MT increases drastically. It indicates that the model trained on short music sequences cannot well generalize to long sequences (line 246-250). In addition, our ablation study shows that both the coarse-grained attention and the bar selection benefit the overall performance, so we cannot say bar summary is more fundamental.
>
> **Q4: How is model on a dataset like Maestro?**
>
> We conduct the experiments on Maestro. Museformer can still exceed all the baseline models in the objective evaluation. Subjective evaluation shows it can generate comparable music with other models.
>
> **Q5: The demos have very different instrumentation and beginning. Are all these samples generated from scratch?**
>
> Yes, they are generated from scratch. Since there are many different styles of music with different instrumentation in the training set, plus sampling is used in generation, it is normal for the variety.

---

> > ### Comment · Reviewer_MUNC · 2022-08-07
> > **Response to the authors**
> >
> > Thank you for your response.
> >
> > **W1, Q1: The assumption about music structure may not be general for all music styles.**
> >
> > Thank you for providing the more detailed statistics. Still, it is a too strong assumption because a statistical pattern shows only the distribution but cannot account for each data sample. According to my observation of the histogram, at least 1/3 of the structure information cannot be captured by the 2**n assumption. For me, it is okay to assume binary hierarchical structure to music to carry on a study. But such assumption is not a solution to reveal the complicated music structure.
> >
> >
> > **W3: Figure 6 is not well-formatted. The example is not musical enough.**
> >
> > Yes, it looks great.
> >
> > **W4, Q6: Demos are not musical enough.**
> >
> > Thank you for pointing this out. It's true that the successful modeling whole song structure and inter-track relation can be a real contribution. If so, I would suggest writing some discussion, sample study, or at least objective measurement in the paper to prove the claim. A strong prove would be for whole-song structure, some evidence that the model is not only structured in "binary hierarchical structure" but also some coherency and developments; for inter-track relation, some example of perfect instrumentation as in voice range assignment, chord voicing, and counterpoint etc.
> >
> >
> > **W5: The writing is a bit wordy and sometimes explains things illogically.**
> >
> > **Q2, W2: In table 2 and the relevant text, what test is used and how is p-value computed?**
> >
> > Yes. Thank you.
> >
> > **Q3, W2: Are "Museformer w/o coarse-grained attention" and Music Transformer almost the same? If so, why PPLs are so different? Is "bar summary" more fundamental?**
> >
> > Sorry, I guess I meant the difference between "Museformer w/o structure-related bar selection" and Music Transformer, as the former only has coarse-grained attention (attended to every summary) and latter is attended to every token. Comparing Table 3 and Table 1, it seems introducing "bar summary" ("Museformer w/o structure-related bar selection" v.s. Music Transformer) gives a fundamental improvement, but "structure-related bar selection" is only marginal? Please correct me if I am wrong.
> >
> > **Q4: How is model on a dataset like Maestro?**
> >
> > Would you include that to your paper? Now I see the authors believe in the model is strong at generating piano solos, capturing inter-track relation and whole-song structure. I would suggest the author summarize and prove the strong points in the paper in a clearer and more convincing way.
> >
> > **Q5: The demos have very different instrumentation and beginning. Are all these samples generated from scratch?**
> >
> > Thank you for clarification.

---

> > > ### Author Response · Authors · 2022-08-09
> > > **Response to Reviewer MUNC**
> > >
> > > Due to the character limits, our response is divided into two comments. This is the first half.
> > >
> > > ---
> > >
> > > Thanks for the new response.
> > >
> > > **W1, Q1: The assumption about music structure may not be general for all music styles.**
> > >
> > > Music structures can be diverse, so it is quite hard to fit for each data sample. However, our structure-related bar pattern has already covered most music (pop, jazz, symphony, etc), as presented on [this page](https://museformer.github.io/rebuttal). This fact should already make our method a practical solution of generating songs with reasonable structures.
> > >
> > > Also, please note that Museformer can encode not only the music structures within the selected bars, but also those outside the selected bars, so it does not work only when the bar pattern fits for the sample. This can be demonstrated by: 1) As presented [here](https://museformer.github.io/rebuttal#music_analysis), compared to the baselines, the similarity distribution of Museformer’s demos is very close to that of the training data not only on the structure-related bars but also on other bars. 2) In the demo 3 video on the [demo page](https://museformer.github.io/), at 1:33, as we annotated, the music piece repeats a previous piece that is 40 bars away, even if 40 is not in the selected bar pattern. The ability to encode information outside the selected bars should be accredited to the following two facts: 1) The receptive field expands when multiple layers of Museformer are stacked up. For example, if the first layer can directly attend to the previous 4th bar, and the second layer can directly attend to the previous 1st bar, then the second layer should be able to indirectly capture the information from the previous 5th bar. In practice, we use a multi-layer architecture, with each layer attending to the selected 8 bars. The accumulated receptive field of the layers should cover a large range and encode the music structures in other bars. 2) In addition to fine-grained attention, our coarse-grained attention captures the summarized information of the other bars. Therefore, although the proposed bar selection for the data sample should better the music structure modeling, our method does not strictly depend on the bar selection to function well.
> > >
> > > Finally, we would like to reiterate that, the general idea that to combine fine- and coarse-grained attention is universal and flexible to all music, and the manually selected bar pattern can already work well for most music, so we believe our work is a valuable exploration. We cannot really require this first attempt to completely solve the structure problem, and future exploration may focus on automatic relevant bar detection. In fact, we have been working on selective attention mechanism to achieve it, for which this work provides an important basis.
> > >
> > > We will add the corresponding discussion to our paper. Thank you so much for providing this opinion that makes our paper better!
> > >
> > > **W4, Q6: Demos are not musical enough.**
> > >
> > > Thanks, and it is really a good suggestion.
> > >
> > > We have presented the similarity distribution statistics over demos generated by different models at [here](https://museformer.github.io/rebuttal#music_analysis), and the self-similarity matrices of the demos as well as the analysis at [here](https://museformer.github.io/rebuttal#ssm), which can be the objective measurement and discussion for the whole-song structure. These materials show that our method can generate both repetitions and variations, with the similarity distribution similar to the training data. The structures can aslo appear in bars other than the selected ones, as we have described in detail in the above response to “W1, Q1”.
> > >
> > > As for the multi-instrument modeling, our presented demos show reasonable instrumentation. Take demo 1 as an example, it has 5 instruments. The voice range assignment is valid for each instrument. The melody is the same for bar 9-16 and bar 25-32, and for these two pieces, the chord progression is also the same. However, the drumset plays more densely in the later piece, showing the development of music compared to the former piece. The multiple instruments bring two challenges: 1) The number of tokens increases drastically compared to single-track music, which results in the difficulty of modeling the whole-song structures. 2) The interplay among instruments. The proposed FC-Attention enables Museformer to efficiently model long sequences, which solves the first challenge. The interplay problem is indeed a valuable problem to solve, and we will leave it for future work.
> > >
> > > As you suggested, we will add this discussion as well as more detailed analysis to the paper.

---

> > > ### Author Response · Authors · 2022-08-09
> > > **Response to Reviewer MUNC**
> > >
> > > Due to the character limits, our response is divided into two comments. This is the second half.
> > >
> > > ---
> > >
> > > **Q3, W2: Are "Museformer w/o coarse-grained attention" and Music Transformer almost the same? If so, why PPLs are so different? Should we conclude "bar summary" is a more fundamental improvement compared to FC-attention?**
> > >
> > > Actually, "Museformer w/o structure-related bar selection" does not mean that it only has coarse-grained attention (bar summary). It means that the selected bars for fine-grained attention are the most recent 8 bars, instead of the structure-related bars elaborately selected by us, i.e., the 1st, 2nd, 4th, 8th, 16th, 24th, 32nd bar. Therefore, the differences between "Museformer w/o structure-related bar selection" and Music Transformer are: 1) the former only directly attends to the recent 8 bars, while the latter directly attends to all the previous tokens; 2) the former has coarse-grained attention (bar summary) while the latter does not. So, the comparison between them cannot result in the conclusion that bar summary gives a fundamental improvement. However, since the former is better than the latter, we can draw the concolusion that the combination of the fine- and coarse-grained attention is an effective replacement to the full attention. It also decreases the complexity and enables the model to encode long sequences. That the PPLs are so different and the PPL of Music Transformer increases drastically when the length increases is because, as we explained in the paper at the baseline introduction part, Music Transformer trained on chunked sequences cannot well predict the music tokens when the sequence length is larger than the chunk size, which indicates that directly applying the model trained on short sequences cannot well generalize to long sequences, and using a long-sequence model like Museformer to generate full-song music is a better choice.
> > >
> > > **Q4: How is the model on a dataset where we have less track but more complicated? (E.g., a dataset like Maestro)**
> > >
> > > Thank you for your suggestion. We presented the results on Maestro 3.0 on [this page](https://museformer.github.io/rebuttal#maestro_results). Museformer exceeds the baseline models on both objective and subjective evaluation. However, as you may see, the ratings of short-term and long-term structures for all the models are relatively low. In recent days, we have further looked into the Maestro dataset to analyze the reasons, and observed that the dataset is relatively low-quality and is lack of music structures that prevail in most music: 1) The size of the dataset is small (only 1700+ songs), which is insufficient for training deep learning models. 2) What is more important is that, according to the dataset introduction [here](https://magenta.tensorflow.org/datasets/maestro), the MIDIs are direct performance recordings rather than calibrated standard sheet music. Since human performers usually do not strictly follow their sheets plus different performers may have different performance styles like rubato, the note onsets and durations can drastically shift, which makes the structures much more complex to be accurately modeled. We display at [here](https://museformer.github.io/rebuttal#maestro_analysis)  a screenshot of a MIDI sample and the similarity distribution of the dataset to demonstrate our claim. Therefore, we do not think Maestro is a proper and representative dataset to demonstrate our contribution of modeling full-song music structures, and thus we may not include it to our paper.
> > >
> > > ---
> > >
> > > We sincerely thank you for providing us so many suggestions that have made our paper better. Love.

---

### Meta-Review · Area_Chair_U5hV · 2022-08-26

**Recommendation:** Accept
**Confidence:** Less certain

**Metareview:**

This paper presents a novel attention selection scheme for modeling the long-term structure of symbolic music. Modeling multi-track music with repetitive structure is a challenging task, especially due to the quadratic memory problem of the transformer. The authors propose measure-limited fine-grained attention and coarse-grained attention that attends to summarizations of measure, not individual tokens. The experiments show that the proposed method can achieve better perplexity compared to the other transformer variants in multi-track symbolic music generation.

One of the main questions raised by the reviewers was the musical quality of the generated examples, such as how the repetitive structure was generated or how it would work in other types of dataset. The authors responded with many additional results that can answer those questions. Though one of the reviewers was not fully satisfied with the evaluation, which is valuable criticism that can improve this paper, the majority of reviewers have agreed that the paper has a clear contribution to be introduced in NeurIPS.

## Strengths
- The paper proposes a nicely working solution for the widely-known but critical limitation of long-sequence modeling of a transformer. The idea is clear, and the implementation is sound.
- The experiment result and the presented generation examples show that the proposed attention strategy can help model long symbolic music sequences.

## Weaknesses
- The evaluation lacks musical analysis of the generated results. It is complicated to evaluate the quality of generated music samples only using the NLL and listening tests with limited examples and subjects. After the reviewer's feedback, the authors presented additional results, including repeat statistics and self-similarity matrices of each track. Since the criticized weaknesses can be largely improved by these additional results, I hope the authors can figure out a good solution to include these contents somehow in the revised paper.
- Even though it can be applied to many kinds of music, the selection of structure-related bar (1,2,4,8, 16) is a strong assumption. It is a reasonable choice if one has to select it rule-based, but there is a clear limitation of this approach, as the authors also agreed.

## Recommendation
- BP-transformer for natural language has a similar concept to the proposed coarse-grained attention in terms that each token attends to the summary of far-located tokens.
  - Ye, Zihao, et al. "BP-Transformer: Modelling long-range context via binary partitioning." arXiv preprint arXiv:1911.04070 (2019)
- As one of the reviewers criticized, a detailed analysis of generated examples is necessary. The authors might consider asking music experts to do a qualitative analysis on the generated examples. Current subjective evaluation has many limitations as a musical background of subjective can be very important to evaluate the quality of music structure. Even though it would be difficult to include those detailed analyses in the main paper, it would be beneficial for the authors and readers to understand the characteristics of Museformer truly. The following paper is a good example of analyzing machine-generated music.
  - Sturm, Bob L., and Oded Ben-Tal. "Taking the models back to music practice: Evaluating generative transcription models built using deep learning." Journal of Creative Music Systems 2 (2017): 32-60.
- The result also suggests the possibility that coarse-grained bar-level summarization is more important than fine-grained attention with structure-related bars. It is not clear which part is more essential to model long-term structures. Therefore, I recommend authors do an ablation study not only just with perplexity score but with detailed analysis of the actual generated examples from two different models (coarse-grained only, and fine-grained only).

**Award:**

No

---

### Decision · Program_Chairs · 2022-09-14

Accept